# Metabolomics to Exploit the Primed Immune System of Tomato Fruit

**DOI:** 10.3390/metabo10030096

**Published:** 2020-03-06

**Authors:** Estrella Luna, Amélie Flandin, Cédric Cassan, Sylvain Prigent, Chloé Chevanne, Camélia Feyrouse Kadiri, Yves Gibon, Pierre Pétriacq

**Affiliations:** 1School of Biosciences, Uni. Birmingham, Birmingham B15 2TT, UK; 2UMR BFP, University Bordeaux, INRAE, 33882 Villenave d’Ornon, France; 3Bordeaux Metabolome, MetaboHUB, PHENOME-EMPHASIS, 33140 Villenave d’Ornon, France

**Keywords:** tomato, metabolomics, biochemical phenotyping, priming, BABA, *Botrytis cinerea*, *Phytophthora infestans*, *Pseudomonas syringae*

## Abstract

Tomato is a major crop suffering substantial yield losses from diseases, as fruit decay at a postharvest level can claim up to 50% of the total production worldwide. Due to the environmental risks of fungicides, there is an increasing interest in exploiting plant immunity through priming, which is an adaptive strategy that improves plant defensive capacity by stimulating induced mechanisms. Broad-spectrum defence priming can be triggered by the compound ß-aminobutyric acid (BABA). In tomato plants, BABA induces resistance against various fungal and bacterial pathogens and different methods of application result in durable protection. Here, we demonstrate that the treatment of tomato plants with BABA resulted in a durable induced resistance in tomato fruit against *Botrytis cinerea*, *Phytophthora infestans* and *Pseudomonas syringae*. Targeted and untargeted metabolomics were used to investigate the metabolic regulations that underpin the priming of tomato fruit against pathogenic microbes that present different infection strategies. Metabolomic analyses revealed major changes after BABA treatment and after inoculation. Remarkably, primed responses seemed specific to the type of infection, rather than showing a common fingerprint of BABA-induced priming. Furthermore, top-down modelling from the detected metabolic markers allowed for the accurate prediction of the measured resistance to fruit pathogens and demonstrated that soluble sugars are essential to predict resistance to fruit pathogens. Altogether, our results demonstrate that metabolomics is particularly insightful for a better understanding of defence priming in fruit. Further experiments are underway in order to identify key metabolites that mediate broad-spectrum BABA-induced priming in tomato fruit.

## 1. Introduction

The increase in world food demand and the indiscriminate use of chemical fertilisation highlight the need to adopt sustainable crop production strategies. Given the major threat of phytopathogenic microbes to food production [1] and ecosystem stability worldwide [2], novel practices are needed to combat these threats. Tomatoes are a highly consumed fruit that represent the eleventh largest commodity, with nearly 183 million tons produced in 4 million hectares in 2017 [3]. Crop yields are strongly affected by filamentous and bacterial pathogens, including the fungus *Botrytis cinerea*, the oomycete *Phytophthora infestans* and the bacterium *Pseudomonas syringae.* These pathogens can claim the complete loss of the crop within days of exposure [4,5,6]. Currently, strategies of control against these biological threats are based on the use of chemical pesticides applied at a pre-harvest stage. However, up to 50% of tomato losses occur at a post-harvest stage [7] due to, among many reasons, the inability to use chemicals at this stage due to residue toxicity. Therefore, new methods of disease control are needed in order to control infections by pathogenic microbes. Exploiting the plant immune system can represent an effective strategy to provide sustainable disease protection [8,9].

Plants are able to defend themselves against pathogens thanks to their innate immune system [10]. In addition, plants are able to sensitise their immune system to protect themselves against biotic threats. This is known as the priming of defence, which is commonly referred as the adaptive part of the plant immune system [11]. Priming occurs after the perception of stimuli that lead to an enhanced responsiveness of defence mechanisms upon subsequent attack [12]. Among stimuli, the chemical agent β-aminobutyric acid (BABA) has been widely studied for its capacity to result in broad-spectrum-induced resistance (IR) in a broad range of plant species [13]. BABA is a non-protein amino acid that has been demonstrated to be a plant product [14]. The work done in the model plant *Arabidopsis* (*A.*) *thaliana* revealed that this outstanding performance is due to the priming activity of multiple signalling pathways [13]. BABA primes salicylic acid (SA)-dependent defences and the deposition of callose, which result in effective protection against biotrophic and necrotrophic pathogens, respectively [15,16]. Importantly, it has been reported that, in many plant species, treatments with BABA result in a stress phenotype that manifests as changes in plant development (e.g., growth, yield, seed production) [17,18,19]. The discovery of the molecular receptor of BABA in *A. thaliana* sheds light into the reasons behind the stress response associated with this chemical: BABA binds to an aspartyl-tRNA synthetase and blocks the enzyme, consequently triggering the accumulation of its canonical substrate, uncharged tRNA, which leads to the activation of the stress response associated with amino acid imbalance in the plant [20]. Moreover, high concentrations of BABA or in specific plant species such as potatoes, can also lead to stress, as the chemical directly activates defence mechanisms, which is a costly trade-off in terms of energy resources to the plant [19,21].

In the tomato system, BABA is known to be able to induce resistance against at least 10 different pests and pathogens, including *B. cinerea*, *P. infestans* and *P. syringae* [13]. Similarly to what has been described in *A. thaliana*, priming of SA-dependent mechanisms has been reported [13]. However, it is likely that further priming mechanisms are responsible for its capacity to induce broad-spectrum resistance. Moreover, studies have demonstrated that priming of BABA is long-lasting [22]. For instance, it has been reported that, after the treatment of tomato seedlings, BABA-IR against *B. cinerea* is maintained for weeks in leaves [18]. Moreover, analysis of the resistance phenotypes in tomato fruit demonstrated that BABA-IR reaches the fruiting stages, as tomatoes from plants that had been treated with BABA at the seedling stage were more resistant to *B. cinerea* than the control [23].

Treatments of plants with BABA did not impact yield or fruit size but resulted in delayed fruit production and ripening [23]. In fruit, durable induced resistance has been linked to the accumulation of specific metabolites. For example, it was reported that, after treatment of seedlings with BABA, there was an accumulation of metabolites associated with alkaloid, terpenoid or jasmonate pathways [23]. It was therefore speculated that these metabolites could be responsible for the enhanced resistance, and therefore could mark the priming fingerprint in tomato fruit [24]. Importantly, however, the metabolites responsible for expression of priming of defence mechanisms after infection still remain unknown. Here, we aimed to determine the metabolic shifts that underpin the BABA priming of immune responses against pathogens of a different nature that infect fruit, including necrotrophic and biotrophic microbes. A combination of targeted biochemical phenotyping for major plant compounds involved in central metabolism and untargeted metabolomics could unveil discriminant metabolic biomarkers that respond to BABA priming and inoculations. To the best of our knowledge, this is the first metabolomic study with multiple fruit pathosystems in relation to the priming of immune responses.

## 2. Results

### 2.1. Effect of BABA on Broad-Spectrum Resistance in Fruit

We determined whether the treatment of tomato plants with BABA resulted in a durable induced resistance in tomato fruit against the fungal necrotrophic pathogen *B. cinerea* (*Bot*), the biotrophic oomycete *P. infestans* (*Phy*) and the hemibiotrophic bacteria *P. syringae* pv. *tomato* (*Pst*). Box-plots showing sample datapoints and mean of the scored symptoms (*n* = 11) revealed that BABA-treated plants produced fruit that were statistically more resistant to *Bot*, *Phy* and *Pst* (*P* < 0.01) than the controls (Figure 1). This indicates that BABA is able to induce resistance in tomato plants for a durable fruit protection against pathogens that have different infection strategies.

### 2.2. Effect of BABA on Fruit Yield and Development

We investigated whether the treatment of tomato seedlings with BABA could impact fruit development and yield. As described previously [23], no differences were found in fruit size (Figure 2A), but delayed fruit production (Figure 2B) and fruit ripening (Figure 2C,D) were reported. However, it was observed that, at 13 weeks of growth, the proportion of green fruit was statistically significantly higher in BABA-treated plants (40%) compared to control plants (33%) (Figure 2D), indicating that fruit production did not slowdown in BABA-treated plants after ripening processes had begun (Figure 2D). This might suggest a positive trade-off for BABA treatment on fruit development.

### 2.3. Global Metabolomics after BABA Treatment and After Inoculation

In order to substantiate the disease resistance after BABA treatment (Figure 1), we investigated fruit metabolism from ethanol extracts of freeze-dried tomato pericarps (*n* = 4) using *i*) targeted biochemical profiling of several major compounds involved in the central metabolism [25,26], and *ii*) untargeted metabolomics of semi-polar metabolites, including specialised compounds, via ultra-high-performance liquid chromatography coupled to electrospray ionisation orbitrap high-resolution mass spectrometry (thereafter referred to as LCMS). For this, the first, second and third fruit developed in the plants were used for pathoassays with *B. cinerea (Bot)*, *P. infestans (Phy)* and *P. syringae (Pst)*, respectively. Unbiased processing of LCMS data, followed by filtering of the most reliable variables, generated 6887 metabolomic features (see Materials and Methods). A global overview of metabolic profiles was visualised by an unsupervised multivariate statistical method, Principal Component Analysis (PCA), for all combinations of priming and pathosystems (Figure 3).

Firstly, for the 6887 metabolomic signals (Figure 3A), PCA explained 35% of the maximal variance of the dataset and resulted in a clear differentiation of the water and BABA treatments, thus suggesting a greater impact on metabolomic profiles for direct BABA application as compared to other conditions. This was confirmed by a univariate statistical method through a two-factor ANOVA (*P* < 0.05), which quantitatively resulted in more statistically significant markers for the BABA factor (3052; 44%) than for the inoculation factor (2309; 33%) or the interaction (401; 6%) (Table 1).

Secondly, we tested 10 major compounds involved in central metabolism (sucrose, fructose, glucose, starch, fructose-6-P, glucose-6-P, glutamate, malate, fumarate and total proteins), as well as total polyphenols. PCA explained 83% of the maximal variance in the dataset and resulted in a separation of fruit by developmental characteristics (i.e., the first and second fruit versus the third fruit) rather than by pathosystems (Figure 3B). Hence, this multivariate differentiation indicates that the profiles of primary metabolites mostly respond to the developmental stage of the fruit, which supports the idea that central metabolism is tuned to fruit growth [27,28,29]. Complementarily, two-factor ANOVA (*P* < 0.05) only generated significant markers for the inoculation factor (4; 36%), including sucrose, fructose, glutamate and fumarate (Table 1 and Appendix A). Interestingly, such markers dropped upon *Bot* and *Phy* infections, while they were not drastically affected by *Pst* infection (Appendix A). Besides, fructose pools remained low across all treatments within the *Pst* pathoassay (i.e., third fruit). Altogether, this indicates that fruit infection affects the pools of central metabolites and those changes depend on the pathosystem.

Furthermore, a partial segregation of pathogen inoculations was observed on the PCA score plots obtained for each pathosystem from a dataset combining LCMS and targeted analyses (Figure 4). This was further exemplified by a supervised Partial Least Square Discriminant Analysis (PLS-DA) allowing a better differentiation of pathosystems and priming treatments (Appendix A). Two-factor ANOVA (*P* < 0.05) for each pathosystem not only confirmed that the BABA factor quantitatively outweighed the inoculation factor and the interaction, but also showed that all these factors were substantial (Table 2). Hence, this indicates that microbial challenges elicit distinct metabolic profiles. Overall, these results reveal metabolic shifts in fruit upon BABA priming and after pathogen inoculation, notably towards semi-polar biochemicals potentially involved in plant stress mitigation (i.e., specialised metabolites).

### 2.4. Primed Responses to Specific Pathogenic Microbes

To gain more insight into BABA priming upon different fruit infections, we next performed quantitative binary comparisons of metabolic markers for each pathosystem by comparing water-treated, mock-inoculated fruit versus *i*) BABA-treated, mock-inoculated fruit, *ii*) water-treated, pathogen-inoculated fruit, and *iii*) BABA-treated, pathogen-inoculated fruit. The resulting statistically significant metabolic markers (*t*-test, *P* < 0.01) were used to construct Venn diagrams showing common and specific markers (Figure 5A). Very few overlaps were observed between BABA (red) and pathogen (blue) conditions (2, 2 and 0 for *Bot*, *Phy* and *Pst*, respectively) and between pathogen and BABA priming (green) conditions (7, 4 and 4 for *Bot*, *Phy* and *Pst*, respectively). Instead, several markers overlapped between BABA treatment and BABA priming (118, 12 and 158 for *Bot*, *Phy*, and *Pst* respectively), and most markers were found either for BABA treatment (*Phy* and *Pst*) or for BABA priming (*Bot*) (Figure 5A). This suggests that BABA results largely in the accumulation of metabolites that could be used during the expression of priming. In addition, metabolic markers that specifically responded to BABA priming in the different fruit pathosystems were compared through a Venn diagram in order to reveal the common metabolic signatures of BABA priming against the three different pathogens (Figure 5B). Strikingly, no common markers were found upon the three infections, although very few markers were observed between *Pst* and *Bot* (10), *Pst* and *Phy* (3), and *Bot* and *Phy* (1). Hence, the primed responses are likely tailored to the encountered pathogenic microbes.

### 2.5. Putative Annotation of Metabolic Markers

We then conducted a tentative annotation of the 14 metabolic markers that were common to *Pst* and *Bot* (10), *Pst* and *Phy* (3), and *Bot* and *Phy* (1) based on their detected *m/z* by high-resolution orbitrap-MS (Table 3). A Kruskal–Wallis test with correction for false rate discovery (Benjamini–Hochberg, *P* < 0.05) confirmed that these 14 metabolic markers showed statistically significant variations that were visualised by bar charts (Appendix A). Putative prediction of compounds and pathways indicated several markers that belonged to the plant defence metabolism, including stress hormones and flavonoids. Interestingly, fungal pathogens (*Bot*, *Phy*) were associated with the induction of the putative marker jasmonoyl–isoleucine. Besides this, (hemi)biotrophic microbes (*Pst* and *Phy*) triggered the accumulation of putative salicylic derivatives and flavonoids (Appendix A). The *Pst*-related primed response further correlated with the depletion of a putative cytokinine. Hence, our results suggest that BABA priming against three different fruit pathogens rely on the induction of pathways involved in the defence hormonal metabolism. Further analytical studies are required to confirm the putative annotation of these priming markers.

### 2.6. Modelling of Resistance to Multiple Fruit Pathogens

Using a predictive biology approach based on generalised linear models [30], we aimed to determine whether resistance to fruit pathogens could be predicted by the detected metabolic markers (Figure 6). Based on those models (Figure 6A), good correlations were observed between measured and predicted values (mean = 0.87), and were statistically different from correlations based on randomly generated resistance (*t*-test, *P* < 2.2 × 10^−16^), which indicated the robustness of the predictions. Furthermore, according to the occurrence of metabolic markers in the models (Figure 6B), fructose appeared to be the best positively correlated predictor (appearing in 99% of the models), as well as sucrose, to a lesser extent (appearing in 41% of the models) (Appendix A). Most predictors (32 out of 34) also showed a high statistical significance from a Kruskal–Wallis test with correction for false rate discovery (Benjamini–Hochberg, *P* < 0.05, Appendix A). This corroborates the outcome from the two-way ANOVA method (Table 1 and Table 2, and Appendix A). Hence, this indicates that soluble sugars involved in the central metabolism are essential to predict resistance to fruit pathogens. The analysis of such compounds is therefore critical for studies involving fruit–pathogen interactions. In addition to sugars, other metabolic predictors appearing in more than 25% of the models showed positive (19 markers) and negative correlations (15 markers) (Figure 6B and Appendix A). Further analytical studies are required to annotate and/or identify such markers. Nonetheless, a tentative annotation of the top 15 predictors based on their detected *m/z* by high-resolution orbitrap-MS is presented in Appendix A. Unsurprisingly, the resulting putative metabolites belonged to defence pathways (i.e., phenolics, flavonoids, terpenes, amino acid conjugates) and lipids. This suggests that immune perception and signalling seem pivotal in predicting resistance to fruit pathogens.

## 3. Discussion

In the present study, we evaluated the metabolic composition of tomato fruit in relation to the BABA-priming of young tomato plants and the infection of three different pathogens at the fruit stage. To the best of our current knowledge, this is the first metabolomic study on three different fruit pathosytems interacting with BABA priming.

Firstly, untargeted metabolomic profiling indicated a great impact of BABA treatment on metabolic profiles (Figure 3A and Table 1). Hence, the treatment of young tomato plants with BABA metabolically primes fruit tissues, and this stimulation was likely more critical than it was for the pathogen inoculations. This might result from the hormonal nature of BABA, which deeply affects plant metabolism, or from stress-related responses that are activated by the chemical itself, as it has been reported previously for high concentrations of BABA or in another *Solanum* species (i.e., potato) [21].

Secondly, targeted analyses of compounds involved in central metabolism demonstrated that the primary metabolic pools responded to the pathosystem inoculations, which reflected the developmental stage of the fruit, as exemplified by the multivariate distinction between the first/second fruit and the third fruit (Figure 3B). In complement, fructose, sucrose, fumarate and glutamate showed statistically significant variations upon inoculation (Table 1). Since fruit of slightly different ages harbour different profiles of primary compounds, we could assume that central metabolism is tuned to fruit growth, more specifically soluble sugars, amino and organic acids. This agrees with previous phenotyping and modelling studies on tomato that demonstrate metabolic shifts in carbon metabolism in the growing fruit [25,28,29,31]. Furthermore, it has been recently confirmed through transcriptomics and proteomics that the developing fruit not only undergoes metabolic shifts in central pathways, but also redox metabolism, such as for pyridine nucleotides that are detrimental to energy homeostasis [27,32,33]. However, major questions remain regarding the nature and dynamics of shifts in central metabolism upon pathogen inoculation. It is reasonable to expect that further investigations involving a more comprehensive view of fruit primary metabolism and how microbial challenges dynamically affect such pathways might significantly improve our understanding of the relationships between central metabolism and fruit–pathogen interactions. In turn, this should provide novel strategies to obtain fruit of better quality and stress resilience [32].

Upon pathogen challenge, while BABA is effective in leaf tissue, very little is known about its contribution in fruit. According to our fruit pathoassays (Figure 1), the treatment of tomato seedlings with BABA resulted in a broad-spectrum resistance against microbes that have different infection strategies, including necrotrophic or biotrophic, and fungal, oomycete or bacterial pathogens. Further, the primed responses are tailored to the encountered pathogen, as exemplified by the little overlap between the different primed states of the three pathosystems (Figure 5). This implies that the induced resistance state is very specific, which strongly suggests that BABA primes multiple signalling pathways through which such different microbes are resisted in the fruit. Among those metabolic responses, hormonal regulations appear detrimental to BABA-induced immunity [23,24,34]. Accordingly, putative annotation of metabolomic markers indicates that hormone conjugates, including salicylic and jasmonic derivatives, and other defence compounds (i.e., flavonoids), are induced upon infection and BABA treatment (Table 3). Given the diverse set of immune responses that the fruit deploys against different microbial stresses, our study highlights the adaptability of priming as a “stimulus-dependent plasticity of response traits” [35]. For this reason, the exact underlying molecular mechanisms of priming are difficult to describe precisely and their description requires further research [36].

Whilst BABA treatment in many plant species results in a stress phenotype that manifests through developmental alterations (e.g., growth, yield, seed production) [17,18,19], we found no differences upon BABA application in fruit size, but observed delayed fruit production or fruit ripening (Figure 2), as described previously [23]. However, after the number of ripened fruit had equalised between both treatments, BABA-treated plants continued producing fruit at a much faster rate than the water-treated plants (Figure 2). Seemingly, through its induction of immune responses, BABA thus provides a positive fitness element for tomato plants. This trade-off might emerge, in part, from the stimulation of various signalling pathways, more specifically the ones that link to the central metabolism, such as amino acids or carbohydrates [34]. As a result, BABA-treated plants would perform particularly well. This agrees with what we know about plant perception of BABA in *A. thaliana*. The binding of BABA by an aspartyl-tRNA synthetase blocks the enzyme, consequently triggering the accumulation of its canonical substrate, uncharged tRNA, which leads to changes in amino acid pools in the plant, therefore affecting primary metabolism [20]. Subsequent signalling modulations might result from an alteration in amino acid precursors (e.g., ethylene, auxin) that would alter fruit production [37].

Despite its economic importance, the molecular mechanisms underlying the pathogenicity of *B. cinerea, P. infestans* and *P. syringae* are poorly understood in fruit. From a computational systems biology perspective, the study of plant–pathogen interactions involved structural and comparative genomics, transcriptomics, and protein–protein interactions [38]. Further, high-resolution metabolome data and sufficient datapoints over time are essential to calculate metabolite coefficients and thus predict metabolic fluxes [39]. Recently, genome-scale metabolic models of *Solanum* species (i.e., potato, tomato) and *Phy* have been integrated to simulate the metabolic fluxes that occur during infection [40,41]. These studies yield insights into the molecular aspects of photosynthesis suppression by *Phy* via the flux of carboxylation to oxygenation reactions, or the nutrient intakes by *Phy* during different phases of the infection cycle. Interestingly, stage-specific profiles embedded in the joint metabolism of the host and pathogen could potentially be refined by integrating the high-resolution metabolome data of tomato infection [41]. Such elegant works involve leaf tissues. Here, we show that fruit metabolomics and modelling can assist in addressing fruit–pathogen interactions. Using top-down modelling based on the construction of generalised linear models [30], we demonstrate that metabolomics data can be used to accurately predict the measured resistance to various fruit pathogens (Figure 6A). Besides, through the evaluation of the occurrence of best predictors, our data indicate that soluble sugars, more specifically fructose [42], and defence metabolites are pivotal to predict the resistance to fruit pathogens (Appendix A). Clearly, a more global systems biology approach based on a higher level of variation in the conditions (e.g., multiple genotypes or priming treatments, various growth stages of the fruit, several infection points) will shed some light on the underlying mechanisms of fruit–pathogen interactions.

Overall, our study validates the value of metabolomics and modelling approaches in the field of phytopathological investigations. This work provides a great perspective for the structural elucidation of the key metabolites involved in broad-spectrum BABA-induced priming in tomato fruit. Although it was only possible to tentatively annotate metabolic biomarkers on the basis of detected HR-accurate *m/z* (Table 3 and Appendix A), our analytical and statistical approach can be further optimised for, e.g., metabolite identification through structural elucidation by NMR or targeted MS/MS analyses. A combination of LCMS with purification steps (e.g., SPE cartridge, fractionation) could prove useful for de novo identification.

## 4. Materials and Methods

### 4.1. Tomato Cultivation

Tomato (*Solanum lycopersicum*) Micro-Tom was used for all experiments described in this publication. Seeds were incubated for 4 days in wet paper at 28 °C to promote homogeneous germination. Germinates were then planted in individual 80 mL pots containing M3 soil. Plants were grown in a controlled-environment greenhouse chamber with 16h of light, at 26 °C, and 8 h of darkness at 21 °C, and 200 µM.m^−1^.s^−1^ light intensity. Experiments were performed from November 2017 until May 2018 in the United Kingdom.

### 4.2. Biochemicals, Reagents and Treatments

All solvents and reagents used in this study were of analytical or MS grades. Β-aminobutyric acid (BABA) was obtained from Sigma-Aldrich (A4420-7). Treatments with BABA were performed entirely as described in [23]. Briefly, 2 week-old tomato seedlings were soil-drenched with 8 mL per pot of either water or 5 mM BABA solution, to generate a final concentration of 0.5 mm in the soil. One week post-treatment, roots were carefully washed under running tap water and then plants were transplanted into individual 2.2 L pots containing untreated M3 soil. Plants were allowed to grow for between 9 and 12 weeks until the fruit turned red, at which point they were harvested and infected with the different pathogens.

### 4.3. Fitness Parameters of Tomato Fruit

Growth and yield were assessed entirely as described in [23]. Assessment of fruit ripening was done as described in [29], by classifying fruit in different levels of maturity by colour.

### 4.4. Pathogens and Inoculations

Cultivations of *Botrytis cinerea* strain R16 [43], *Phytophthora infestans* 88,069 [44] and *Pseudomonas syringae* pv. *tomato DC3000* (*Pst DC3000*) [45] were done as described in the corresponding publications. For inoculations with *B. cinerea,* the first fruit were used. Inoculations were performed entirely as described in [23]. For infections with *P. infestans,* the second fruit were used. Inoculations were performed by placing 10 µl drops of a spore concentration of 5 × 10^4^ spores/mL onto the needle-wounded tip of the tomato fruit. After infection, fruit were kept at 20 ºC in the dark. For *P. syringae* infections, the third fruit were used. Inoculations were done by spraying bacteria onto the fruit in a concentration of 10^8^ cells/mL in 10 mM MgSO_4_ and 0.05% (v/v) Silwet L-77. Infected fruit were kept in the dark at 25 °C. Mock inoculations were performed by following the exact same protocols but without pathogens in the solutions. Fruit were 56 days post-anthesis (dpa), 63 dpa and 70 dpa for the first, second and third fruit, respectively.

Scoring of *B. cinerea* symptoms were performed entirely as described in [23]. Scoring of *P. infestans* disease was done by classifying lesions into different categories of fruit colonization: Class 0; healthy, Class I; necrosis associated with the lesion, Class II; necrosis and mycelium associated with the lesion, Class III; necrosis and mycelium spread in the fruit. Scoring of *Pst DC3000* disease was done by classifying lesions into different categories of fruit damage: Class 0; healthy, Class I; turgent but cracking fruit, Class II; cracked fruit, Class III; fruit tissue collapse. Disease severity rates were calculated from the nominal lesion categories of four fruit per plant (*n* = 11), as described in [46]. Statistical analysis of disease phenotypes was performed as described in [23].

### 4.5. Metabolite Extraction

For metabolome analysis, the first, second and third fruit developed by plants were used for pathoassays with *B. cinerea (Bot)*, *P. infestans (Phy)* and *P. syringae (Pst)*, respectively. Experiments on each type of fruit were separated by one week. Infections were performed as described above when the corresponding fruit were fully ripened. Two days after inoculation with the different pathogens, fresh pericarps were rapidly collected into 2 mL-microtubes, then flash-frozen in liquid nitrogen and freeze-dried for 72 h (Pilote Compact, SARL CRYOTEC, Saint-Gély-du-Fesc, France). Fine grinding of dried material was subsequently performed using a ball mixer for 2 min at 30 Hz (Retsch Mill MM400, fisher scientific, Bordeaux, France) after adding two metal beads (Beads inox AISI 400C 5 mm, CIMAP, Caen, France) to each tube (Micro-tube, 2 mL PP, Sarstedt, Germany). Ten milligrams of each replicated sample were weighed into 1.1 mL-micronic tubes (MP32033L, Micronic, Lelystad, Netherlands), randomised onto a 96-micronic rack (MPW51001BC6, Micronic, Lelystad, Netherlands) then capped using a robotised capper–decapper (Decapper 193000/00, Hamilton, Bienne, Switzerland). Each rack also contained an empty tube corresponding to the extraction blank. The resulting micronics were then stored at −80 °C. Extraction of metabolites was conducted on four biologically replicated pericarp samples (*n* = 4) using a robotised extraction method developed at *Bordeaux Metabolome Facility* (https://metabolome.cgfb.u-bordeaux.fr/en, Villenave d’Ornon, France). The robot was a bespoken piece of equipment that allowed for pipetting solvents, mixing, cooling and centrifuging racks of micronics. After decapping the micronics, the extraction began by adding 300 µL of solvent A containing 80% ethanol and 0.1% formic acid (v/v) with 250 µg/mL methyl vanillate as the internal standard. Racks were agitated on the robot (30 sec, 500 rpm) then placed for 15 min into a sonicator containing ice-cold water (Elmasonic S300, Elma, Singen, Germany). Racks were put back on the robot and centrifuged (5 min, 1350 *g*). The first round of extraction stopped by pipetting 300 µL of the resulting supernatant into new 1.1 mL-micronic tubes. A second round of extraction was performed with 300 µl of solvent A, and the resulting pellet was finally washed with solvent B (50% ethanol (v/v)). The micronics-containing supernatants were kept for filtration and the micronics with the pellets were kept for further starch and total protein analysis.

Filtration was also robotised (Microlab STARlet, Hamilton, Bienne, Switzerland) and allowed for the transfer of the supernatants onto a filtration 96-well sterile clear plate (MSGVS2210, 0.22 µM, Hydrophil. Low Protein Binding Durapore, Millipore, Molsheim, France) according to the supplier’s instruction. Filtrates were subsequently collected into a new micronic tube. Finally, quality control (QC) samples were prepared by robotically pipetting 15 µL of each sample into a single tube that was mixed afterwards (Microlab STARlet, Hamilton, Bienne, Switzerland). Each rack was supplemented with a micronic tube containing the QC mix. The QC sample was replicated six times along the project run.

### 4.6. Targeted Biochemical Phenotyping

Targeted analyses of sucrose, fructose, glucose, starch, fructose-6-P, glucose-6-P, glutamate, malate, fumarate, total soluble proteins and total polyphenols were conducted on the HiTMe plateau at *Bordeaux Metabolome Facility*. Measurements were based on coupled enzyme assays as described previously [25,26], except for total soluble proteins that were evaluated via Bradford assay [47], and total phenols that were measured colorimetrically using a redox reaction with Folin–Ciocalteu reagent and gallic acid as the standard [48].

### 4.7. Untargeted Metabolic Profiling

Untargeted metabolic profiling by UHPLC-LTQ-Orbitrap mass spectrometry (LCMS) was performed using an Ultimate 3000 ultra-high-pressure liquid chromatography (UHPLC) system coupled to an LTQ-Orbitrap Elite mass spectrometer interfaced with an electrospray (ESI) ionisation source (ThermoScientific, Bremen, Germany). The system was controlled by Thermo XCalibur v.3.0.63 software. Chromatographic separation was achieved at a flow rate of 350 µL/min using a GEMINI UHPLC C18 column (150 × 2 mm, 3 µm, Le Pecq, Phenomenex, France) coupled to a C18 SecurityGuard GEMINI pre-column (4 × 2 mm, 3 µm, Le Pecq, Phenomenex, France). The column was maintained at 35 °C and the injection volume was 5 µL. The mobile phase consisted of solvent A (0.05 % (v/v) formic acid in water) and solvent B (acetonitrile) with the following gradient: 0–0.5 min 3% B, 0.5–1 min 3% B, 1–9 min 50% B, 9–13 min 100% B, 13–14 min 100% B, 14–14.5 min 3% B, 14.5–18 min 3% B. Ionisation of samples was performed in both negative and positive mode with the following parameters: ESI^-^ (Heater temp: 300 °C, Sheath Gas Flow Rate: 45 (arb), Aux Gas Flow Rate: 15 (arb), Sweep Gas Flow Rate: 10 (arb), I Spray Voltage: 2.5 kV, Capillary Temp: 300 °C, S-Lens RF Level: 60%), and ESI^+^ (Heater temp: 300 °C, Sheath Gas Flow Rate: 60 (arb), Aux Gas Flow Rate: 20 (arb), Sweep Gas Flow Rate: 10 (arb), I Spray Voltage: 3.2 kV, Capillary Temp: 300 °C, S-Lens RF Level: 55%). MS full scan detection of ions was operated by FTMS (50–1500 Da) at a resolution of 240,000. Prior to analyses, the LTQ-Orbitrap was calibrated by infusing a solution of the calibration dependent of the ionisation mode (Pierce© ESI Negative Ion Calibration Solution (ref: 88324); Pierce LTQ Velos ESI Positive Ion Calibration solution (ref: 88323). The injection sequence started with three blank extracts, then three QC samples, then one blank extract, and each group of samples was subsequently injected, followed by a blank extract. Another two QC samples were injected throughout the analysis. In total, six QC samples and 16 blank extracts were injected to correct for mass spectrometer signal drift, and to filter out variables detected in blanks, respectively.

### 4.8. Processing and Statistical Analysis of Metabolomic Datasets

Processing of raw LCMS data using XCMS in R (v 3.6.1) [49] yielded 10,875 detected RT-*m/z* pairs for ESI^+^ and 5,796 for ESI^-^. After data-cleaning (blank check, Δ_RT_ < 60 s, Δ*_m/z_* < 0.015 Da, CV QC < 30%), 6887 variables were retained for further chemiometrics. Both untargeted and targeted metabolomic data were first normalised by median normalisation, cube-root transformation and Pareto scaling using MetaboAnalyst v.3 [50] before applying multivariate and univariate statistical analyses [51]. The normalised dataset is available as Appendix A. PCA and PLS-DA were performed with MetaboAnalyst v.3 providing satisfactory validation parameters of the multivariate models (R^2^ > 0.87 and Q^2^ > 0.35). PC coordinates for metabolomic features that are responsible for PC1 and PC2 are presented in Appendix A. Univariate statistical methods were performed using MeV v.4.9.0. [52] at *P* < 0.05 for two-factor ANOVA and *P* < 0.01 for binary comparisons by *t*-tests. In addition, MarVis v 1 was used to confirm the statistically significant variation om the priming markers through a Kruskal–Wallis test at *P* < 0.05 with correction for false discovery rate [53,54]. Putative annotation of such markers was performed by screening the detected exact *m/z* against multiple online databases, including METLIN chemical database (https://metlin.scripps.edu/) [55] and KNApSAcK (http://kanaya.naist.jp/KNApSAcK/) [56]. The resulting predicted pathways were checked using the PubChem database (https://pubchem.ncbi.nlm.nih.gov/).

### 4.9. Top-down Modelling Approach

Generalised linear models were constructed in R (v 3.6.1) using the *glmnet* package (v 3.0-2) [30] in order to identify potential links between detected metabolic markers and resistance to the fruit pathogens. Those models were used to predict resistance values based on the detected metabolic markers. Cross-validation was applied by randomly dividing the datasets into two parts: 80 % of the individuals were used to construct the models and 20 % to check for the quality of the prediction. The quality of the models was assessed based on the mean square error between real and predicted values. To cope with this randomisation, 500 models were constructed for each measurement. Generalised linear models contain a penalisation value, allowing less informative variables to be discarded as this value increases (1000 values were tested for each of the 500 models), hence variables occurring the most in the models can be seen as the most stable predictors of resistance to biotic challenges. Given the high number of metabolic variables and the relatively small set of plants, 500 randomly generated resistance datasets were created to estimate the chances of predicting random values. A Student’s *t*-test was used to compare the quality of predictions of real and random resistances. 

## Figures and Tables

**Figure 1 metabolites-10-00096-f001:**
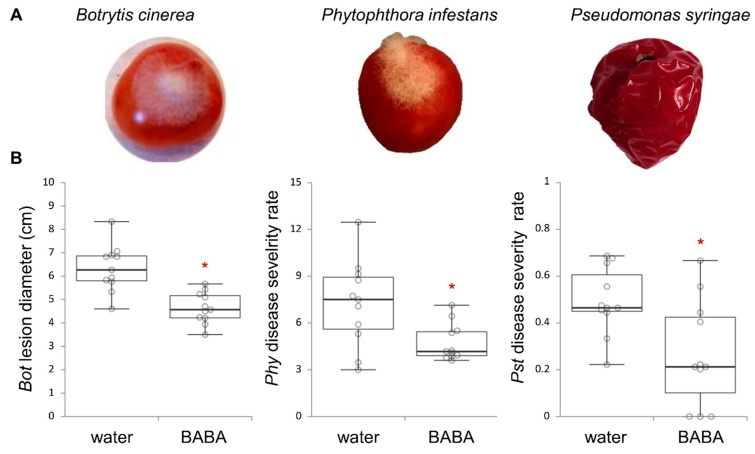
ß-aminobutyric acid (BABA) primes tomato fruit for a durable disease resistance against three different pathogenic microbes. (**A**) photographs showing tomato fruit 2 days after infection with *Botrytis cinerea* (*Bot*, left), *Phytophthora infestans* (*Phy*, middle) or *Pseudomonas syringae* pv. *tomato* (*Pst*, right). (**B**) box-plots of disease symptoms from fruit that originated from plants treated with water or BABA (500 µM) then inoculated with water as a mock control or with *Bot* (left), *Phy* (middle) and *Pst* (right). Symptoms were scored 2 days after inoculation from 11 biologically replicated fruit (*n* = 11). Asterisks indicate statistically significant differences between water- and BABA-treated plants *(t-*test, *P* < 0.01).

**Figure 2 metabolites-10-00096-f002:**
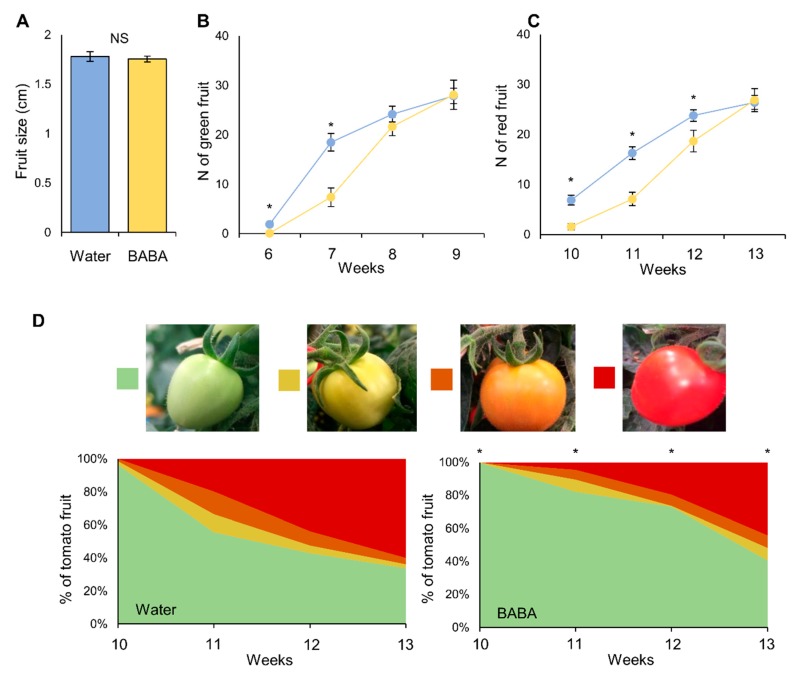
Effect of BABA on fruit yield and development. (**A**) fruit size (cm) from plants treated with water (blue) or 500 µM BABA (yellow). NS: not statistically significant (*t*-test, *P >* 0.05). (**B**) number of green fruit produced from water (blue)- or BABA (yellow)-treated plants after weeks of growth. Asterisks indicate statistically significant differences (*t*-test, P > 0.05). (**C**) number of ripped fruit produced from water (blue)- or BABA (yellow)-treated plants after weeks of growth. Asterisks indicate statistically significant differences (*t*-test, *P <* 0.05). (**D**) Proportion of fruit at different stages of fruit ripening in control and BABA-treated plants, expressed as percentage of occurrence per treatment at different weeks of growth. Asterisks indicate statistically significant differences between distributions at specific timepoints (Chi-square test, *P <* 0.05).

**Figure 3 metabolites-10-00096-f003:**
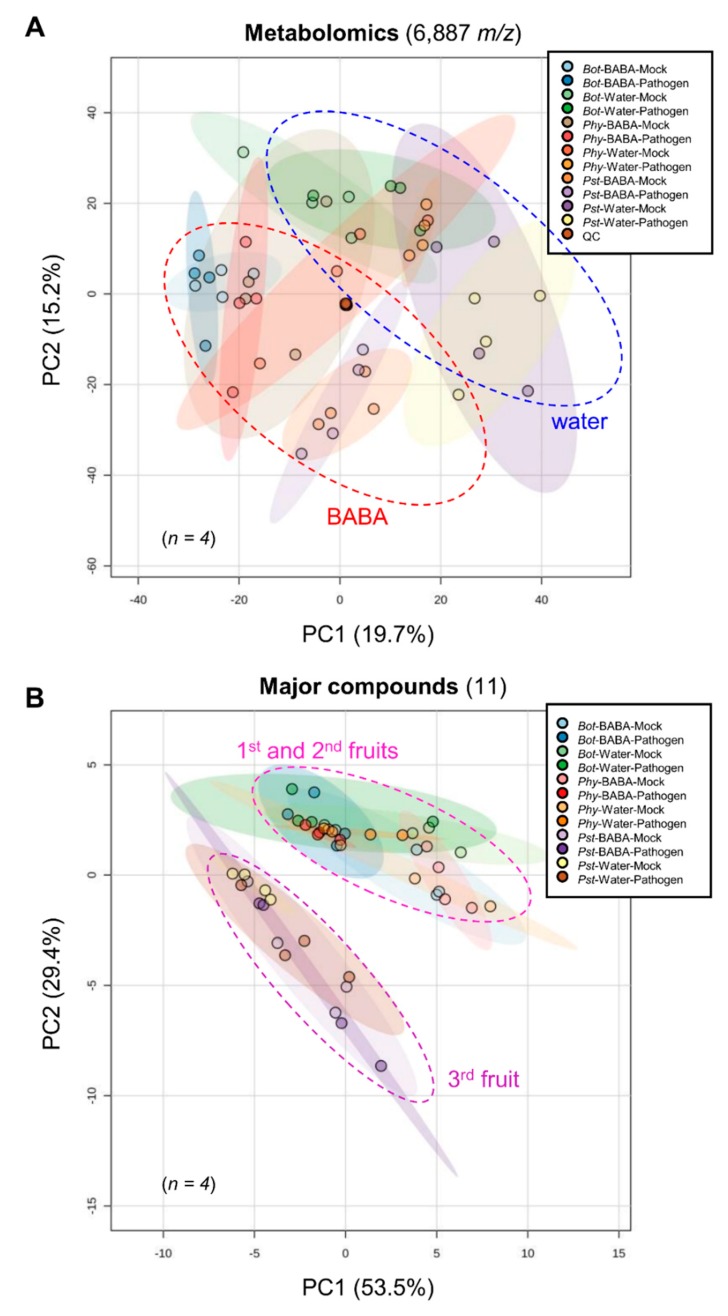
Global metabolomic changes after BABA treatment and after pathogen inoculation. Principle component analysis (PCA) score plots (*n* = 4) of 6887 LCMS-based metabolomics features (**A**) and 11 major compounds analysed by targeted biochemical phenotyping (**B**). Maximal variance explained by each PC is given in brackets.

**Figure 4 metabolites-10-00096-f004:**
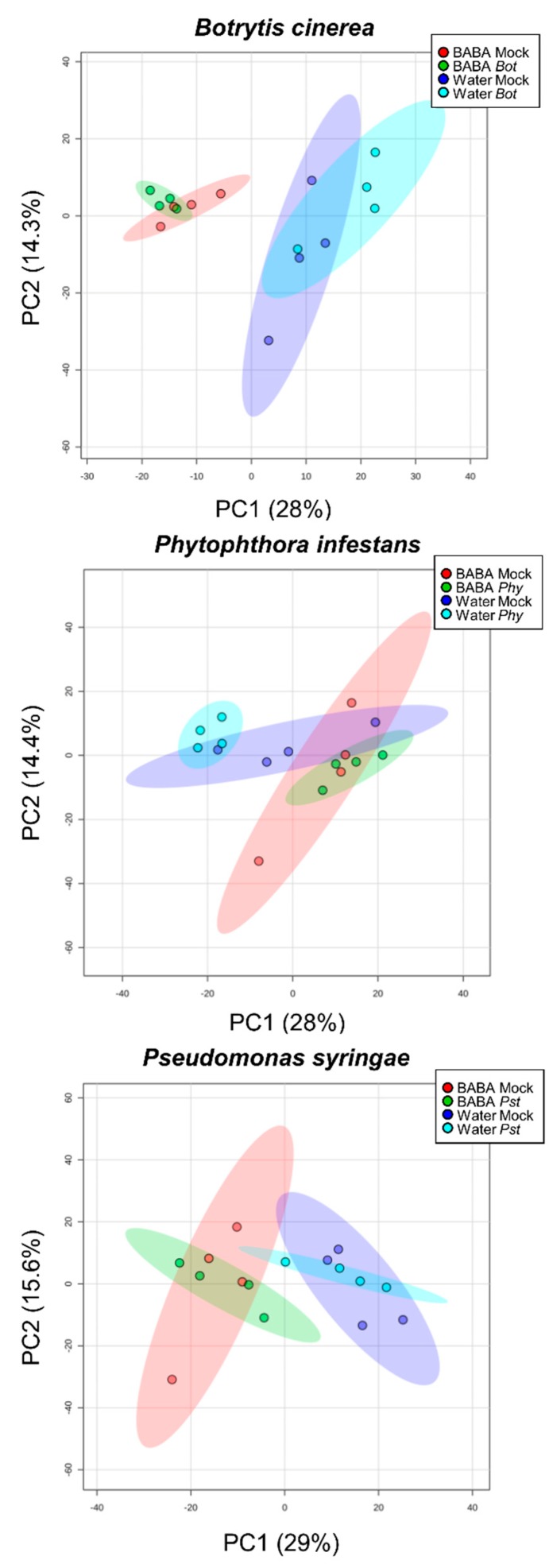
Partial segregation of pathogen inoculations for each fruit pathosystem. PCA score plots (*n* = 4) of 6898 features (6887 electrospray ionisation orbitrap high-resolution mass spectrometry (LCMS) variables + 11 major compounds) showing metabolomics overview between the three different pathosystems. Maximal variance explained by each PC are given in brackets.

**Figure 5 metabolites-10-00096-f005:**
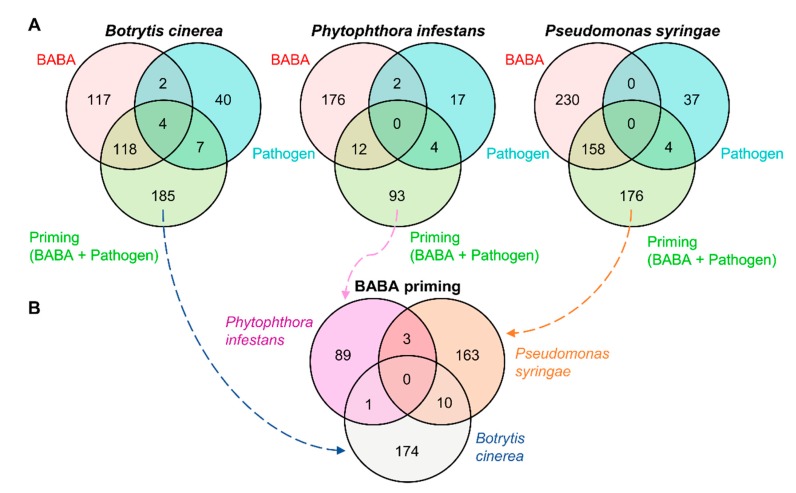
The primed responses are tailored to the encountered pathogenic microbes. (**A**) Venn diagrams showing quantitative binary comparisons were performed for each pathosystem (*t*-test, *n* = 4, *P* < 0.01) between water-treated, mock-inoculated fruit versus BABA-treated, mock-inoculated fruit (BABA, red); water-treated, pathogen-inoculated fruit (pathogen, blue); BABA-treated, pathogen-inoculated fruit (priming, green). (**B**) Venn diagrams showing the resulting priming clusters for each fruit pathosystem.

**Figure 6 metabolites-10-00096-f006:**
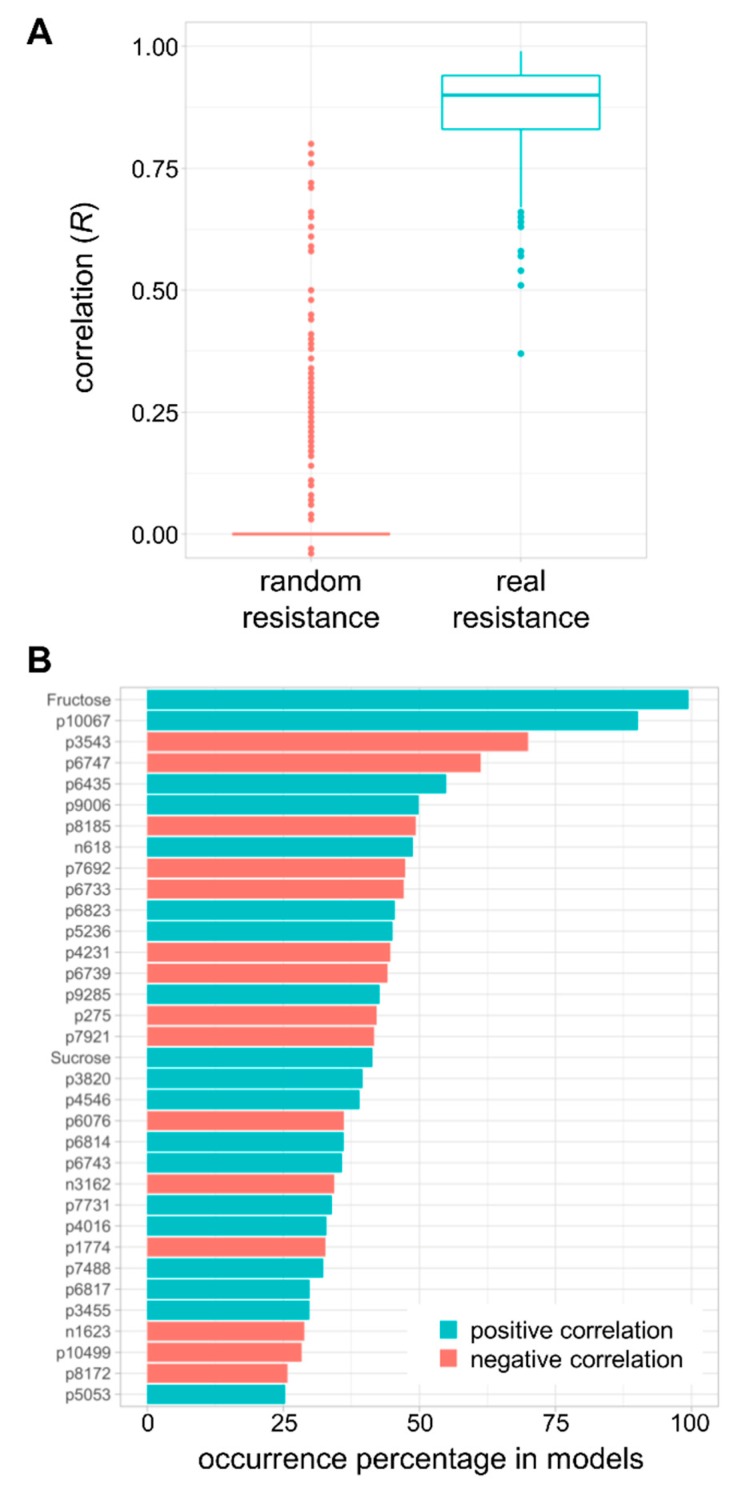
Prediction of biotic resistance from metabolic markers. (**A**) correlation between predicted and measured resistance based on generalised linear models. (**B**) occurrence (%) of the metabolic markers in the models that showed a positive or negative correlation with the resistance to fruit pathogens. Details of untargeted markers are presented in Appendix A.

**Table 1 metabolites-10-00096-t001:** Univariate statistical analysis of the metabolomic features and major compounds.

Two factors ANOVA (*P* < 0.05)		BABA (Water vs. BABA)	Inoculation (Mock vs. *Bot/Phy/Pst*)	BABA × Inoculation
Metabolomic features	Total 6887	3052	2309	401
Major compounds	Total 11	0	4	0

**Table 2 metabolites-10-00096-t002:** Univariate statistical analysis for each pathosystem.

Two Factors ANOVA (*P* < 0.05)		BABA	Inoculation	BABA × Inoculation
		Water vs. BABA	Mock vs. Pathogen	
*Botrytis cinerea*	Total 6998	2297	724	482
*Phytophthora infestans*	Total 6998	1728	805	674
*Pseudomonas syringae*	Total 6998	2097	322	313

**Table 3 metabolites-10-00096-t003:** Putative annotation of the primed response markers.

Primed Response	Detected *m/z* (Da) ^1^	RT (min) ^1^	P Value ^2^	ESI Mode	Putative Adduct	Predicted *m/z*	Δppm	Putative Compound	Predicted Formula	Putative Pathway
*Bot* and *Phy*	346.1975	4.1	5.3 × 10^−3^	+	[M+Na]+	323.2097	4	Jasmonoyl-isoleucine	C18H29NO4	Jasmonates
*Bot* and *Pst*	450.1119	7.5	6.7 × 10^−3^	-	[M+F]-	431.1206	15	Ribosylzeatin phosphate	C15H22N5O8P	Cytokinines
	289.0896	2.5	5.6 × 10^−3^	+	[M+H-2H_2_O]+	324.0998	8	5,6-Dimethoxy-[2’’,3’’:7,8]furanoflavanone	C19H16O5	Flavonoids
	380.1489	3.6	5.3 × 10^−3^	+	[M+ACN+H]+	338.1154	0	3,5,7-Trihydroxy-6-prenylflavone	C20H18O5	Flavonoids
	451.1238	7.4	8.4 × 10^−3^	+	[M+H]+	450.1162	0	3,4,2’,4’,6’-Pentahydroxychalcone 2’-glucoside	C21H22O11	Flavonoids
	437.2213	7.1	3.8 × 10^−2^	+	[M+ACN+Na]+	373.1889	37	Jasmonoyl-tyrosine	C21H27NO5	Jasmonates
	535.3117	9.1	1.6 × 10^−2^	-	[M-H]-	536.3114	14	Phosphoglycerolipid (20:2(11Z,14Z)/0:0)	C26H49O9P	Lipids
	268.2271	5.4	2.5 × 10^−2^	+	[M+NH4]+	250.1933	0	C16:3n-6,9,12	C16H26O2	Lipids
	652.4048	6.2	3.7 × 10^−2^	+	[M+NH4]+	634.3870	24	3-trans-p-Coumaroyl-rotundic acid	C39H54O7	Phenolics
	272.0644	6.0	1.4 × 10^−2^	-	[M+F]-	253.0586	27	Salicyloyl-aspartic acid	C11H11NO6	Salicylic derivatives
	154.0216	6.0	1.2 × 10^−2^	+	-	-	-	-	-	Unknown
*Pst* and *Phy*	479.1402	4.3	4.2 × 10^−2^	-	[M+Na-2H]-	458.1577	16	7-Hydroxy-5,4’-dimethoxy-8-methylisoflavone 7-O-rhamnoside	C24H26O9	Flavonoids
	525.1456	4.2	2.6 × 10^−2^	+	[M+NH4]+	507.1139	3	Delphinidin 3-(acetylglucoside)	C23H23O13	Flavonoids
	452.1954	4.0	8.2 × 10^−3^	-	[M+CH3COO]-	393.1940	27	Diphenhydramine salicylate	C24H27NO4	Salicylic derivatives

^1^: metabolomic parameters detected by LCMS. ^2^: *P* indicating the statistical significance from a Kruskal–Wallis test followed by correction for false discovery rate using the Benjamini–Hochberg method.

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
