# Peer review of "Metabolomics to Exploit the Primed Immune System of Tomato Fruit"

_metabolites, 2020, doi:10.3390/metabo10030096_

Round 1
Reviewer 1 Report
The manuscript by Luna et al. provides new data on the effects of BABA priming on Botrytis cinerea, Phytophthora infestans and Pseudomonas syringae infection on severity of diseases and metabolic changes in tomato fruits. The study is original and is based on an extremely high number (6,887) of LCMS-based metabolomics features and the concentration of 11 major compounds identified by targeted analysis. The results can be of interest of readers working in the research field of plant metabolomics. Nevertheless, there are a few aspects of the study which are not clear for me from the manuscript.
- “The seeds were incubated for 4 days in wet paper at 28oC to promote homogenous germination” (ln327). When were the seeds treated with BABA and for how long time?
- The 1st and 2nd fruits developed, similar in major metabolite composition (Fig. 3b), were infected with Botrytis and Phytophtora, while the 3rd fruits developed, which were different in metabolite composition from the 1st and 2nd fruits (Fig. 3b), were infected with Pseudomonas. How can it be concluded that “primed responses depended entirely on the type of infection, rather than showing a common fingerprint of BABA-induced priming.” (ln28) if the infection was carried out on an ab ovo different metabolite background?
- Which metabolites are responsible for PC1 and PC2 in Fig. 3 and 4?
Minor points:
Fig. 4 and Fig. S2 – All PCA plots are labelled as a Botrytis infection (Bot).
ln247 – Is Table 1 the correct reference?
Fig. S3 - Markers are labelled according to their high resolution detected m/z. Thus, it is difficult to correlate them with the text in section 2.5.
Author Response
Please find below the responses to reviewer 1 in red letter font
1. “The seeds were incubated for 4 days in wet paper at 28oC to promote homogenous germination” (ln327). When were the seeds treated with BABA and for how long time?
As stated in the M&M, BABA treatment was performed as described in Wilkinson et al. 2018 (ref 22). We have complemented the M&M to clarify this point:
Briefly, 2 weeks-old tomato seedling were soil‐drenched with 8 mL per pot of either water or 5 mM BABA solution, to generate a final concentration of 0.5 mm in the soil. One week post‐treatment, roots were carefully washed under running tap water and then plants were transplanted into individual 2.2 L pots containing untreated M3 soil. Plants were allowed to grow for between 9 and 12 weeks until the fruit turned red, at which point they were harvested and infected with the different pathogens.
2. The 1st and 2nd fruits developed, similar in major metabolite composition (Fig. 3b), were infected with Botrytis and Phytophtora, while the 3rd fruits developed, which were different in metabolite composition from the 1st and 2nd fruits (Fig. 3b), were infected with Pseudomonas. How can it be concluded that “primed responses depended entirely on the type of infection, rather than showing a common fingerprint of BABA-induced priming.” (ln28) if the infection was carried out on an ab ovo different metabolite background?
We thank the reviewer for this valuable comment as we appreciate that the three different fruits have different metabolic compositions. Accordingly, we have toned down this sentence in the abstract.
However, we would like to point out to the reviewer that even if the differences seem to be apparent between those 3 fruit, they were located in the same bunch of flowers and therefore created at similar times. This assumption is supported by LCMS data that do not indicate a global metabolomic response with the fruit type (Figure 3a). Moreover, importantly, we are looking at the differences that occur after infection, therefore focusing specifically on the differences in metabolites triggered after the fruit has been infected. It is in that induction where we do not see a common pattern of accumulation and this is why we conclude that primed responses depend on the infection.
3. Which metabolites are responsible for PC1 and PC2 in Fig. 3 and 4?
We cannot provide the metabolites that are responsible for PC1 and PC2 in Fig. 3A and Fig. 4 since the PCA are based on metabolomic features and not metabolites. However, we can provide such information for Fig. 3B based on 11 major compounds. We include in the revised manuscript Supplemental Material 2 with the PC coordinates for the 5 different PCA presented in this study. A sentence has been added in the M&M section.
Minor points:
Fig. 4 and Fig. S2 – All PCA plots are labelled as a Botrytis infection (Bot).
Thanks for checking this. We have replaced the figure inserts with the correct labels.
ln247 – Is Table 1 the correct reference?
Table 1 is the correct reference, highlighting that 4 major compounds show statistically significant changes (fructose, sucrose, fumarate and glutamate).
Fig. S3 - Markers are labelled according to their high resolution detected m/z. Thus, it is difficult to correlate them with the text in section 2.5.
Thanks for the comment. We have added the name of the putative compound in Fig S3.
Reviewer 2 Report
Manuscript ID: metabolites-733243
“Metabolomics to exploit the primed immune system of tomato fruit.”
By Estrella et al.,
The manuscript described, also on the basis of existing literature, the metabolic shifts that underpin BABA priming of immune responses against pathogens of different nature that infect fruit and showed the value of metabolomics and modeling approach in the field of phytopathological investigations.
The manuscript is well written, and the results are very and convincing. However, since it is not widely implemented, this study is of importance to show the effects in multiple fruit pathosystems. The manuscript needs a brief read through explanations of all abbreviations. This manuscript presents an interesting and carefully designed retrospective analysis of BABA control measure. The data were collected and generated in a labor intensive way, followed by a deluge of various statistical analyses. The quality of experimental data is convincing and the conclusions appear to be reliable. I have just a few small comments on the text, which the authors may wish to address:
Results
Lines 106-109 and 113-116: It is not very clear to me the difference between the two treatments shown in figures 1 and 2. I wonder if in the first case they are adult plants and in the second they are seedlings. Please explain this aspect and the meaning. In figure 2 does the infection occurs using the same pests?
Lines 166-181
The Venn diagram shown in Fig. 3 is not very clear. Its meaning is not clear to me. I have a few questions to ask the authors.
More than the common metabolites I would like to know the external ones what do they mean? For example in the first diagram and so on in the others, 117 metabolites are present in Botrytis cinerea when is treated with BABA? I wonder what is their meaning?
Discussion
The discussion is too long, it is repetitive and redundant, up to line 246 it seems a repetition of the results, there are too many speculations and therefore it must be lightened. In my opinion, those altered metabolisms should be emphasized, for example in line 252 the authors report an alteration of the redox metabolism, pyridine nucleotides, it would be interesting to see which metabolites are involved.
Author Response
Please find below the responses to reviewer 2 in red letter font
Lines 106-109 and 113-116: It is not very clear to me the difference between the two treatments shown in figures 1 and 2. I wonder if in the first case they are adult plants and in the second they are seedlings. Please explain this aspect and the meaning. In figure 2 does the infection occurs using the same pests?
Figure 1 shows the different resistance phenotypes of fruit after infection with the 3 different pathogens used in the study: B. cinerea, P. infestans and P. syringae. Figure 2 shows the impact of BABA in fruit production and development so here we have not used any pathogen and we are entirely focusing in phenotypes of stress triggered by BABA. In both figures we are working with fruit rather than with plants and they are all at similar developmental stages.
Lines 166-181: The Venn diagram shown in Fig. 3 is not very clear. Its meaning is not clear to me. I have a few questions to ask the authors. More than the common metabolites I would like to know the external ones what do they mean? For example in the first diagram and so on in the others, 117 metabolites are present in Botrytis cinerea when is treated with BABA? I wonder what is their meaning?
We thank the reviewer for this comment. As exemplified by Venn diagrams, the “external” areas indicate the metabolomic features that are specific to BABA treatment (red), pathogen inoculation (blue) and priming (green) which is achieved by BABA treatment and subsequent pathogen inoculation.
Discussion: The discussion is too long, it is repetitive and redundant, up to line 246 it seems a repetition of the results, there are too many speculations and therefore it must be lightened. In my opinion, those altered metabolisms should be emphasized, for example in line 252 the authors report an alteration of the redox metabolism, pyridine nucleotides, it would be interesting to see which metabolites are involved.
We do appreciate the reviewer’s suggestion to trim the discussion. We have discarded the sections that are not necessary. However, because this is the first study combining BABA priming in fruit and multiple pathosystems, as well as untargeted LCMS and targeted profiling, we do feel that our manuscript needs a thorough discussion to place its importance in the context of phytopathological metabolomics. Line 252, the reference to pyridine nucleotides supports the idea that the growing fruit experiences shits in redox metabolism that links to energy statues. The altered metabolisms are emphasised in Line 246 (fructose, sucrose, fumarate and glutamate), Line 270 (hormones and defence metabolites).